# The Chestnut and the Imperfect Crime: A Case Report of Femicide and Staged Road Accident

**DOI:** 10.3390/diagnostics15212664

**Published:** 2025-10-22

**Authors:** Gennaro Baldino, Tindara Biondo, Cataldo Raffino, Marija Čaplinskienė, Stefano Vanin, Elvira Ventura Spagnolo

**Affiliations:** 1Department of Biomedical and Dental Sciences and Morphofunctional Imaging, University of Messina, Via Consolare Valeria 1, 98125 Messina, Italy; 2Legal Medicine Centre of INAIL, 94100 Enna, Italy; 3State Forensic Science, Mykolas Romeris University Vilnius, 01101 Vilnius, Lithuania; m.caplinskiene@gmail.com; 4Department of Earth, Environmental and Life Sciences (DISTAV), University of Genoa, 16132 Genoa, Italy

**Keywords:** road traffic accident, charred body, femicide, homicide, forensic pathology, multidisciplinary forensic approach

## Abstract

**Introduction:** Charred bodies represents a significant challenge for forensic pathologists due to the destructive effects of fire on human remains. Although most fire-related deaths are accidental, cases of suicide and homicide are not uncommon. **Case Report:** We report a peculiar case of a severely burned body discovered inside a torched vehicle. Under judicial investigation, a full autopsy was performed, including macroscopic and microscopic examination of key anatomical structures: the inspection of oral cavity revealed no soot deposits; a foreign object (a chestnut) was found anterior to the epiglottis, though not lodged within the glottis; no thermal injuries or soot were observed in the upper or lower airways. Histological analysis excluded thermal damage at the alveolar–capillary interface. Alveolar spaces appeared both hyperinflated and ectatic, likely due to septal rupture, suggestive of acute pulmonary emphysema and multiorgan congestion. Carboxyhemoglobin levels were below 5%, indicating a low level which did not support intravital inhalation of combustion gases. Based on the comprehensive medico-legal findings, the cause of death was attributed to an asphyxial mechanism. It was further demonstrated that the burning of the body occurred post-mortem. DNA extraction from two dental specimens enabled positive identification of the victim. Subsequent investigations confirmed the case to be a femicide. The perpetrator, following a domestic altercation over jealousy, suffocated his young wife and attempted to simulate accidental choking by placing a chestnut in her mouth. He then staged a vehicular fire to mimic a fatal accident. **Conclusions:** The case underlines that a multidisciplinary forensic approach is essential, and must integrate different methodologies and the analysis of both circumstantial evidence and scene investigation.

## 1. Introduction

Charred bodies represents a significant challenge for forensic pathologists. The difficulty is primarily due to extensive thermal degradation and charring of human remains, which can obscure the assessment of injuries and pathological findings.

When assessing extensively burned bodies, several forensic issues may arise, including personal identification, assessment of external and internal injuries, securing DNA-evidence, determination of the cause of death, reconstruction of the death scene, and estimation of the post-mortem interval [1]. Although most fire-related fatalities are accidental, causes involving intentional death or the deliberate use of fire after death are also frequently encountered [2,3,4,5,6]. In particular, burning after death can be a deliberate act to destroy evidence of a violent crime, thereby complicating forensic investigation [7,8,9]. In fact, fire can rapidly obliterate biological traces and critical anatomical or environmental markers, significantly hampering forensic reconstruction [10,11,12]. The investigation in such scenarios require a meticulous evaluation of injuries by forensic pathologists, other specialists to generate a multidisciplinary approach. This includes autopsy, radiological imaging, toxicological and genetic analyses, histopathology, forensic odontology, forensic anthropology, and forensic entomology. Additionally, a detailed investigation of the crime scene and a comprehensive review of circumstantial data are essential to analyzing and reconstructing the death scenario with scientific rigor [13,14]. The recovery site survey is a critical, non-repeatable phase in forensic investigation, requiring collaboration between forensic pathologists and law enforcement to ensure proper collection of evidentiary material [2,15,16]. This is especially crucial in cases involving charred remains, where burning may indicate attempts to conceal the body, destroy trace evidence, and hinder identification [2,17].

## 2. Case Report

The case here reported concerns a charred body discovered inside a burned-out vehicle. The body was found in a prone position, with the head and torso resting on the front driver’s seat and the pelvis near the gear shift lever. The vehicle, destroyed by fire, was located approximately 18 m from the roadway at the bottom of a slope and collided with a tree that showed signs of the impact and smoke staining (Figure 1A,B).

Its front end was oriented downhill, with the driver’s side door partially open and the engine compartment hood fully raised, held open by a tree branch. No skid marks or tire tracks were observed on the road asphalt surface. A distinct odor of gasoline was still detectable inside the vehicle. The body was recovered and placed under judicial custody for further forensic examination and autopsy. Postmortem Computed Tomography, with consequent 3D reconstruction, did not indicate any firearm-related injuries, stab or incised wounds, and trauma consistent with blunt force impact. The autopsy revealed that the head was completely charred, with calcination of the cranial bones and destruction of the muscular and cutaneous tissues of both the cranial vault and face. These conditions made it impossible to identify the individual based on physiognomic features. The cranial bones were diastased, with extrusion of brain tissue externally and showing signs of heat-induced coagulative necrosis. No relevant pathological findings were observed on the external surface of the neck or in the underlying musculature (Figure 2A,B).

The upper limbs were carbonized, with complete loss of the forearms and hands. The lower limbs were also carbonized, with absence of the mid and distal thirds of both femurs. Residual tibial segments were present bilaterally, held together by necrotic muscular tissue, with the legs positioned in a flexed attitude over the thighs. The feet were found inside the vehicle, along with fragments of burned clothes. Both anterior and posterior surfaces of the torso were carbonized, exposing fragments of the costovertebral skeletal framework. The abdominal wall was absent, including the skin, subcutaneous tissue, and musculature, resulting in full exposure of the abdominal organs.

During oral cavity inspection, no signs of soot were observed; surprisingly, a foreign body (a chestnut) was observed positioned anterior to the epiglottis, obstructing access to the oropharynx. However, it was not lodged within the glottis (Figure 3A). The object was firm in consistency, intact, without any dental impressions, and weighed 5.6 g (Figure 3B).

A small quantity of soot was found in the proximal tract of the tongue, while its complete absence was noted not only in the lower airways but also in the more proximal regions of the tracheal mucosa. Despite the limitations of the charring, no lesions were found in the hyoid body, thyroid cartilage, or cricoid cartilage. Furthermore, no hemorrhages were detected in the neck. Upon sectioning, the lungs exhibited a diffusely increased and dense consistency, yet retained a clearly recognizable anatomical structure. Foamy secretions were present within the terminal bronchi, with no foreign material observed along the bronchial surfaces. Internal female genital organs were identifiable. Histological analysis revealed no signs of thermal damage at the alveolar–capillary interface. The lung parenchyma exhibited markedly hyperinflated and dilated alveolar spaces, likely resulting from septal disruption, consistent with features of acute pulmonary emphysema (Figure 4). Additionally, evidence of multiorgan vascular congestion was observed. Toxicological analysis of the peripheral venous blood was also performed, detecting carboxyhemoglobin levels in the blood below 5%, a value that is considered toxicologically irrelevant. Screening for alcohol, narcotics, and psychoactive substances in blood samples was negative. The anamnestic and circumstantial data and the negative histological findings for cardiac pathologies (limited by carbonization) did not indicate a natural cause of death. Cause of death was determined to be asphyxia, as evidenced by histopathological findings in the pulmonary tissue. Moreover, the absence of soot within the airways, the lack of thermal lesions in the distal respiratory tract, and the low carboxyhemoglobin levels indicate that the victim was not breathing at the time the body was exposed to fire. Genetic analysis via DNA extraction from two recovered dental elements, enabled the identification of the victim. The perpetrator, possessing technical and investigative expertise, orchestrated a post-homicide scenario intended to simulate accidental death by asphyxiation due to the ingestion of a chestnut. In an effort to eliminate all traces of evidence, the individual subsequently staged a vehicular accident and deliberately set the vehicle on fire. The policy and engineering investigation revealed that the driver’s seat had been positioned excessively rearward, preventing the victim from operating the brake and clutch pedals. The ignition key was found in the “off” position, indicating that at the time of the fire, the engine was not running due to interrupted fuel supply and the vehicle was stationary. The fuel tank was found intact. Based on the absence of skid marks and the nature of the damage sustained by the vehicle, it was determined that the vehicle had been moving at a low speed. These findings led investigators to exclude the possibility that the fire originated from impact with the tree.

## 3. Discussion

The forensic evaluation of extensively burned bodies presents significant diagnostic challenges, as thermal exposure alters or obscures conventional postmortem findings. High temperatures may obliterate pre-existing traumatic lesions or produce heat-induced artifacts that mimic injuries of different etiologies. Thermal effects include cranial vault fractures and segmental disarticulations, such as limb amputations, which can complicate the reconstruction of perimortem events and hinder the determination of the actual cause and manner of death [18].

One of the most important challenges in the forensic examination of charred bodies is determining the vitality of the injuries—specifically, whether the victim was exposed to flames before or after death. When the body surface is not entirely carbonized, external signs such as hyperemic zones (commonly referred to as “red flare”) between burned and intact skin, along with granulation tissue within the lesion, may suggest antemortem exposure. However, as noted by Melez et al. [19], the presence of a red flare alone is not conclusive, as postmortem burns can produce similar findings, leading to false negatives. Autopsy is the most reliable method for confirming whether the victim is alive or dead when a fire has been set. Indicators such as soot deposition in the airways and thermal injury to the respiratory tract support the hypothesis of antemortem exposure. Nonetheless, soot may be absent even in living victims, particularly in cases of rapid combustion (“flash fire”), where death may occur from cardiocirculatory collapse due to extreme heat and rapid oxygen depletion. In the case under investigation, the absence of carbonaceous residues in both superficial and deep respiratory pathways strongly suggests that the victim was already deceased at the time of fire exposure. The presence of soot limited to the distal portion of the tongue is consistent with postmortem deposition. In severely burned bodies, contamination of the upper airways may still occur postmortem, necessitating careful histological evaluation. Histopathological analysis is essential for distinguishing antemortem from postmortem burn lesions. Key indicators of vital reaction include dilated capillaries, coagulative necrosis, nuclear swelling in epidermal cells, subepidermal connective tissue edema, vacuolization, elongation of cells and nuclei, and infiltration of polymorphonuclear leukocytes—all of which are absent in postmortem burns [20,21,22]. In cases of smoke inhalation during life, soot and thermal damage may be observed in the airways. In the present case, histological examination revealed alveolar dilation and panacinar emphysema, with optically empty alveolar spaces and focal hemorrhagic exudate. Acute pulmonary emphysema is a recognized internal marker of asphyxial death, resulting from intense inspiratory and expiratory effort, which may lead to rupture of alveolar septa. Toxicological analysis provides further insight into vitality and the circumstances of death. A blood carboxyhemoglobin (COHb) level exceeding 10%, along with black soot deposits on airway mucosa, indicates respiration during fire exposure and supports antemortem vitality [23], However, the literature notes that chronic tobacco users may exhibit carboxyhemoglobin (COHb) concentrations ranging from 10% to 20%, which complicates the interpretation of such values in the context of fire-related fatalities. Consequently, some authors argue that COHb levels within this range are not definitive indicators of vitality at the time of flame exposure [24]. In contrast, Wirthwein et al. suggest that COHb concentrations exceeding 30% are highly indicative of combustion product inhalation during life, whereas values below 20% should prompt consideration of alternative causes of death, as they may not reliably reflect ante-mortem respiration in fire scenarios [25]. The presence of carbon monoxide (CO) in the bloodstream typically results from inhalation through the respiratory tract, followed by absorption into the circulatory system owing to CO’s high affinity for hemoglobin. This pattern is commonly observed in cases of suicide or accidental death involving fire [26]. Conversely, low or undetectable COHb levels are expected when fire is used postmortem to hide evidence of homicide [27]. In the present case, toxicological analysis revealed no detectable COHb levels in the blood. Regarding the cause of death, a study by Sane et al. [26] identified cranial trauma as the most frequent cause in fire-related homicides, followed by asphyxial death from various mechanisms. The examination of charred remains is complicated by thermal artifacts that may mimic traumatic injuries, such as heat-induced cranial fractures or extradural hemorrhages. Computed tomography (CT) proves highly effective in identifying traumatic lesions obscured by burn damage and in differentiating thermal fractures from those caused by blunt force. Thermal fractures typically exhibit distinct patterns, including longitudinal and transdiploic lines with delamination, whereas traumatic fractures tend to be perpendicular and penetrate deeper into soft tissues [5,28,29,30] CT imaging also assists in distinguishing epidural hematomas of traumatic origin from those resulting from heat exposure. Heat-induced hematomas generally appear crescent-shaped, exhibit low radiodensity, and lack confinement by suture lines—unlike traumatic epidural hematomas, which are lens-shaped, hyperdense, and anatomically bounded by cranial sutures [31]. No single sign of asphyxia can be considered pathognomonic. Asphyxia is classified within the broader category of acute asphyxia syndromes and represents one of five principal subtypes, alongside manual strangulation, hanging, ligature strangulation, drowning and suffocation [32] diagnosis, requiring a multidisciplinary approach that integrates autopsy findings, histological analysis, and investigative context.

In this case, the convergence of all elements derived from autopsy and instrumental findings allowed us to attribute the cause of death—within the bounds of scientific plausibility—to external mechanical asphyxia, likely resulting from homicidal strangulation and/or suffocation. Several alternative scenarios were systematically excluded. First, an accidental death due to a motor vehicle collision followed by fire was ruled out, as was the hypothesis of airway obstruction by a foreign body leading to loss of vehicle control and impact and combustion. Accidental choking was also excluded, given that the foreign object found near the glottis was not impacted; although it may have shifted during the perimortem phase, it was not lodged in a manner consistent with fatal obstruction. PMCT imaging excluded the presence of firearm injuries and sharp force trauma. Blunt force trauma was also ruled out as a cause of death. While thermal effects had caused explosive disintegration of the cranial vault and a subdural hematoma consistent with heat exposure, no vital traumatic injuries were identified. Acute intoxication was excluded based on toxicological analysis, which revealed no presence of alcohol or narcotic substances in the bloodstream. Mechanical asphyxia due to ligature or manual strangulation was also ruled out, as no patterned compression marks were observed on the neck. Given the degree of carbonization, the skin remained macroscopically intact in the cervical region, and no signs of constriction consistent with strangulation were evident. Accurate diagnosis in cases involving charred remains requires evaluation of markers indicative of vitality during flame exposure [12]. Key among these is the presence of soot in the lower respiratory tract, which reflects active inhalation of carbonaceous particles prior to death. Thermal lesions may also be present.

## 4. Conclusions

The case here presented highlights the need for a multidisciplinary forensic investigation of burned human remains, particularly when fire is used as a means to commit or conceal homicide. Postmortem burns—whether inflicted to cause death or to obscure its circumstances—often serve to destroy the victim’s identity, mask the true cause and manner of death, and eliminate physical evidence. Proper management of such cases requires a multidisciplinary approach, integrating findings from forensic pathology, autopsy, postmortem computed tomography (PMCT), toxicological analysis, crime scene reconstruction, and investigative data obtained through police interrogation. In suspected vehicular fires, the presumed manner of death is typically accidental. However, routine forensic protocols must include the exclusion of homicidal or suicidal intent. A thorough autopsy combined with detailed crime scene analysis remains essential for accurately establishing the cause of death.

## Figures and Tables

**Figure 1 diagnostics-15-02664-f001:**
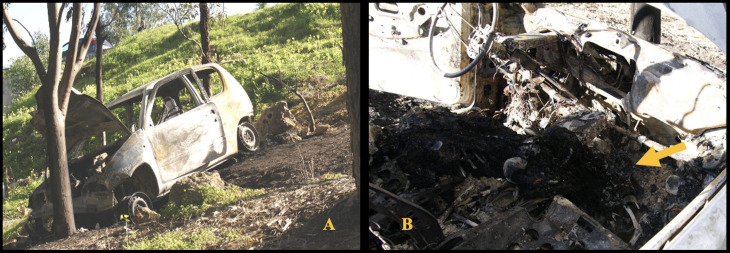
(**A**) Car destroyed by fire. (**B**) Charred body found inside the cabin (yellow arrow).

**Figure 2 diagnostics-15-02664-f002:**
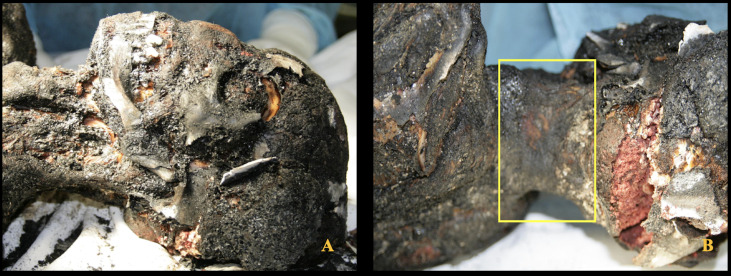
Charred body. (**A**) Anterior region of the neck showing no signs of constriction. (**B**) Posterior region of the neck with partially preserved skin (yellow inset) and no evidence of patterned compressive injuries.

**Figure 3 diagnostics-15-02664-f003:**
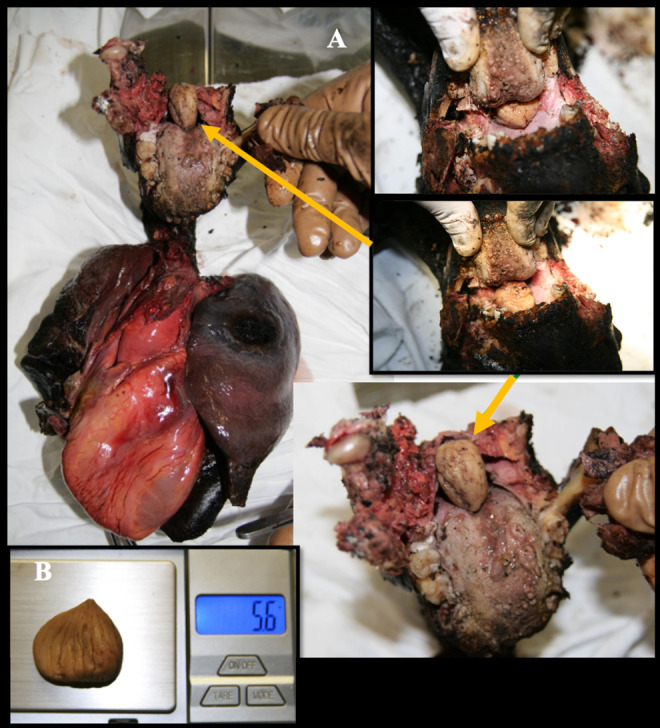
(**A**) Laryngo-tracheo-bronchial axis, heart and lungs: presence of a foreign body (chestnut) free in the oropharynx (yellow arrows), located anterior to the epiglottis and not impacted in the glottis. (**B**) Details of the chestnut, without bite marks.

**Figure 4 diagnostics-15-02664-f004:**
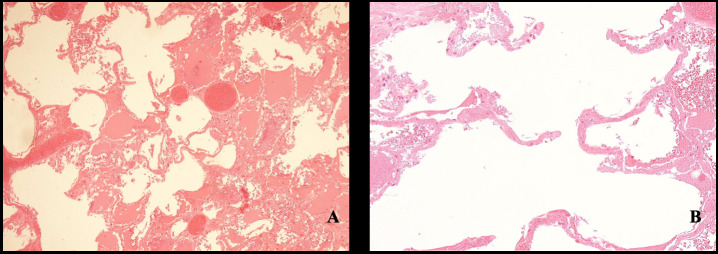
Alveolar parenchyma. (**A**) Congested septa and alveolar dilatation. (**B**) Acute panacinar emphysema.

## Data Availability

All the data are reported in the paper.

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
