# Peer review of "The Chestnut and the Imperfect Crime: A Case Report of Femicide and Staged Road Accident"

_diagnostics, 2025, doi:10.3390/diagnostics15212664_

Round 1

Reviewer 1 Report

Comments and Suggestions for Authors

In the manuscript, the authors present an interesting and original case report of femicide, followed by the concealment of the body through fire. Given the sensitive nature of the topics discussed, this article may be of significant interest to forensic professionals.

The authors also critically analyze the challenges encountered when dealing with charred remains, emphasizing the importance of a multidisciplinary approach to clarify the sequence of events and differentiate between murder, suicide, and accidental death. They appropriately highlight the diagnostic limitations of charred corpses, which are inherently complex.

The case is thoroughly described, with post-mortem analyses (crime scene investigation, autopsy) complemented by toxicological, histopathological, genetic, and PMTC imaging studies. The article is well-supported by relevant figures and an appropriate range of bibliographical references.

Author Response

Thank you for your comments and appreciation of the paper.

Reviewer 2 Report

Comments and Suggestions for Authors

I thank the journal for the opportunity to review the manuscript “The chestnut and the imperfect crime. A case report of femicide and staged road accident”. The manuscript circles around a case report of a staged traffic accident in which a severely burnt body was recovered. The following forensic and policiary investigation ruled that the case was a murder, the fire was staged after the death of the person, to conceal a murder, potentially following suffocation. I find the case report intriguing. However, a case report can also be used for initiating a literature review of a specific finding or circumstance, but this is not performed in the case report at hand. I would have preferred if the authors used the case report as an vignette for some form of review. Nonetheless, the case report by itself is interesting. However, there are some issues I would like to address:

  1. In the introduction it is stated that fatalities from road accidents that leads to fires present significant diagnostic challenges for forensic pathologists. I would suggest that this is the case for most fire fatalities including severely burnt bodies.
  2. Furthermore it is stated that dealing with extensively burnt bodies, several forensic issues may arise, I suggest adding “assessment of external and internal injuries, securing DNA-evidence” to the list of such forensic issues. Furthermore, I would use the word assesseing instead of dealing with.
  3. Line 52, page 2, an “and” is missing from the sentence.
  4. I think that in the introduction there should be more emphasis on the police forensic investigation of both “the accident” site and how the fire started, when listing all experts involved in investigating such a case.
  5. Page 2, line 70 it is stated that PMCT excluded any evidence of firearm-related injury, stab- or incised wounds and trauma associated with blunt force impact. To use the word excluded in this setting is wrong in my opinion, I could agree to use the phrasing – “PMCT did not indicate any fire-arm related injuries …”
  6. For the description of autopsy findings I lack a description if the eyes were still observable and If so, if any petechial bleeds were observed. Were neck structures such as the hyoidbode, thyroid- and cricoid cartilage intact. Were there signs of bleeding in the neck.
  7. As there are little macroscopic findings in the current case, the diagnosis is one of exclusion. A differential diagnosis could be that the persons suffers a heartattack, the car slowly stops in speed and colides with the tree. The fire starts, but the person is already deceased due to the heart attack (I am aware that other circumstances in the case doesn’t support such a scenario, but from the forensic standpoint it is important to have covered it). Were there any diseases observed during the autopsy (such as coronary sclerosis, increased heart weight, pulmonary embolism and so on…).
  8. Are there any information on preexisting health for the deceased – medical charts?
  9. What are the circumstances known in the case, what was the motif?
  10. What did the suspected perpetrator say about the homicidie, how was it performed according to such a statement?
  11. The last part of the discussion is an repetition of what is previously mentioned (page 7, line 234-238)
  12. It is stated that there were no need for informed consent. Is this true also concerning relatives of the deceased – for example the car could be recognized from the included pictures and the geogrpahical area.
  13. In the case report the conclusion to some extent leans up on the acute pulmonary emphysema and the histological assessment thereof – it would be suitable with a discussion on the evidentiary value of these findings for a diagnosis of asphyxia.
  14. In the abstract it is written that “Carboxyhemoglobin levels were below 5 % , indicating minimal exposure to combustion gases” – that I would not say, rather “a low level not in support of intravital inhalation of combustion gases.”
  15. At no point is the age of the deceased individual reported.

Author Response

RESPONSE TO REVIEWERS

Reply to Reviewer 2

I thank the journal for the opportunity to review the manuscript “The chestnut and the imperfect crime. A case report of femicide and staged road accident”. The manuscript circles around a case report of a staged traffic accident in which a severely burnt body was recovered. The following forensic and policiary investigation ruled that the case was a murder, the fire was staged after the death of the person, to conceal a murder, potentially following suffocation. I find the case report intriguing. However, a case report can also be used for initiating a literature review of a specific finding or circumstance, but this is not performed in the case report at hand. I would have preferred if the authors used the case report as an vignette for some form of review. Nonetheless, the case report by itself is interesting. However, there are some issues I would like to address:

We appreciate the reviewers' positive comments and their expressed interest in the case report.

Thank you for your valuable suggestions. We have made the requested changes and additions to the text (highlighted in blue). We also provide responses to the reviewer's questions.

Furthermore, appreciating the reviewer's suggestion, we will prepare a systematic review of the literature on the topic in the near future.

  1. In the introduction it is stated that fatalities from road accidents that leads to fires present significant diagnostic challenges for forensic pathologists. I would suggest that this is the case for most fire fatalities including severely burnt bodies. R.  Thanks for the suggestion. We've changed the sentence in the text.

  1. Furthermore it is stated that dealing with extensively burnt bodies, several forensic issues may arise, I suggest adding “assessment of external and internal injuries, securing DNA-evidence” to the list of such forensic issues. Furthermore, I would use the word assesseing instead of dealing with. R. Thanks for the suggestion; we have modified the text to include " assessment of external and internal injuries, securing DNA-evidence”.

  1. Line 52, page 2, an “and” is missing from the sentence. R. Sorry for the oversight, we put “and” in the sentence.

  1. I think that in the introduction there should be more emphasis on the police forensic investigation of both “the accident” site and how the fire started, when listing all experts involved in investigating such a case. R. Thanks for the suggestion. We've added the data to the text.

  1. Page 2, line 70 it is stated that PMCT excluded any evidence of firearm-related injury, stab or incised wounds and trauma associated with blunt force impact. To use the word excluded in this setting is wrong in my opinion, I could agree to use the phrasing – “PMCT did not indicate any firearm related injuries …” R. Thanks for the suggestion. We've changed the sentence in the text.

  1. For the description of autopsy findings I lack a description if the eyes were still observable and If so, if any petechial bleeds were observed. Were neck structures such as the hyoidbode, thyroid- and cricoid cartilage intact. Were there signs of bleeding in the neck. R. Thank you for your question. Due to the extent of charring, as shown in the images, it was not possible to visualize the eyes or assess the presence of petechial hemorrhages. Regarding the neck structures, despite the limitations imposed by the burn injuries (which nonetheless preserved partial anatomical integrity, as shown in the images), no lesions were observed in the hyoid bone, thyroid cartilage, or cricoid cartilage. Additionally, no hemorrhages were identified in the neck region. We have integrated the information into the revised text.

7. As there are little macroscopic findings in the current case, the diagnosis is one of exclusion. A differential diagnosis could be that the persons suffers a heartattack, the car slowly stops in speed and colides with the tree. The fire starts, but the person is already deceased due to the heart attack (I am aware that other circumstances in the case doesn’t support such a scenario, but from the forensic standpoint it is important to have covered it). Were there any diseases observed during the autopsy (such as coronary sclerosis, increased heart weight, pulmonary embolism and so on…).  8. Are there any information on preexisting health for the deceased – medical charts? R. Thank you for your comments. We agree that in the case of a charred body, such as this one, the diagnosis is one of exclusion. For this reason, it is important to follow a multidisciplinary approach an integrating police investigations, circumstantial data, and forensic medical evidence. In the case we presented both engineering and chemical investigations ruled out vehicle malfunction and excluded the possibility of a self-initiated fire resulting from a technical failure. The heart attack hypothesis was ruled out considering the woman's young age, the negative medical history reported (by family members), and the histological data (although limited by the state of charring of the body). The data have been integrated into the text.

  1. What are the circumstances known in the case, what was the motif? R. Thank you for your question. The femicide by suffocation occurred during a domestic dispute triggered by the husband's well-known jealousy. The information has been added to the text.

  1. What did the suspected perpetrator say about the homicidie, how was it performed according to such a statement? R. He only pleaded guilty and accepted a plea bargain.

  1. The last part of the discussion is an repetition of what is previously mentioned (page 7, line 234-238) R. Thanks for the tip. We apologize for the oversight of the repetition, which has been removed from the text.

  1. It is stated that there were no need for informed consent. Is this true also concerning relatives of the deceased – for example the car could be recognized from the included pictures and the geogrpahical area. R. Also at the publisher's request, the consent point has been integrated into the specific section of the manuscript. In any case, the car and the location are not recognizable.

  1. In the case report the conclusion to some extent leans up on the acute pulmonary emphysema and the histological assessment thereof – it would be suitable with a discussion on the evidentiary value of these findings for a diagnosis of asphyxia. R. Thanks for your comment. We discussed this data in the thread.

  1. In the abstract it is written that “Carboxyhemoglobin levels were below 5 % , indicating minimal exposure to combustion gases” – that I would not say, rather “a low level not in support of intravital inhalation of combustion gases.” R. Thanks for the suggestion, we've changed the sentence in the text.

  1. At no point is the age of the deceased individual reported. R. Thanks for the question. The subject was 34 years old, but we chose not to include that, instead indicating a young woman, also to comply with maintaining anonymity

Round 2

Reviewer 2 Report

Comments and Suggestions for Authors

I thank the authors for the fine amendments to my comments. Unfortunately, there are still a few minor issues to address.

  1. Page 2, line 62: I do not fully understand the sentence”…which can obscure anatomical and pathological indicators” – I suggest…which can obscure the assessment of injuries and pathological findings.
  2. Page 3, line 104: “did not indicative” – should be “did not indicate”.
  3. Page 4, line 153: “allowed us to exclude” – I suggest “did not indicate a natural cause of death”
  4. In the Informed consent statement on page 7, “anonimized” is misspelled, it should be “anonymized”.

Author Response

Reply to Reviewer for Minor Revisions

 Thank you for your observations and comments, which have further helped us improve the article. We've changed the sentences in the text (written in red).

Detailed reply is as follows:

  1. Page 2, line 62: I do not fully understand the sentence”…which can obscure anatomical and pathological indicators” – I suggest…which can obscure the assessment of injuries and pathological findings.   R: Thank you. We've changed the sentences in which can obscure the assessment of injuries and pathological finding” in the text (written in red).
  2. Page 3, line 104: “did not indicative” – should be “did not indicate”.    R: Thank you. We've changed the sentence in did not indicate” (written in red).
  3. Page 4, line 153: “allowed us to exclude”  I suggest “did not indicate a natural cause of death”   R: Thank you. We've changed the sentences in “did not indicate a natural cause of death” (written in red).
  4. In the Informed consent statement on page 7, “anonimized” is misspelled, it should be “anonymized”.   R: Thank you. We modified the word in “anonymized” (written in red).